# Influence of the Synergistic Effect of Multi-Walled Carbon Nanotubes and Carbon Fibers in the Rubber Matrix on the Friction and Wear of Metals during the Mixing Process

**DOI:** 10.3390/polym14183731

**Published:** 2022-09-07

**Authors:** Lin Wang, Yi Pan, Yihui Chen, Jian Qiu, Aihua Du, Deshang Han, Chuansheng Wang

**Affiliations:** 1Key Laboratory of Rubber-Plastics (Ministry of Education), School of Polymer Science and Engineering, Qingdao University of Science and Technology, Qingdao 266042, China; 2College of Electromechanical Engineering, Qingdao University of Science and Technology, Qingdao266061, China; 3National Engineering Research Center of Advanced Tire Equipment and Key Materials, Qingdao 266061, China

**Keywords:** MWCNT/CF/CB composites, spatial mesh structure, wear, the life of the internal mixer

## Abstract

As a piece of high-intensity running equipment, the wear of an internal mixer determines the quality of rubber and its life. In general, the wear of an internal mixer is caused by the friction between the rubber and metal during the mixing process, and the most severe wear position is the end face of the equipment. In this paper, a mixture of multi-walled carbon nanotubes (MWCNTs) and carbon fibers (CFs) are added to rubber by mechanical compounding to obtain MWCNT/CF/carbon black (CB) composites. By investigating the synergistic mechanism of MWCNTs and CFs, we analyze the effect of the MWCNT/CF ratio on the frictional wear of metal on the end face of the internal mixer. At the microscopic level, MWCNTs and CFs form a spatial meshwork with CB particles through synergistic interactions. The CB particles can be adsorbed on the spatial meshwork to promote the dispersion of CB particles. In addition, the formation of oil film can be slowed down due to the spatial meshwork, which could hinder the spillage of aromatic oil. Meanwhile, the spatial meshwork serves as a physical isolation layer between the rubber and metal to reduce friction. Therefore, it dramatically impacts the dispersion degree of CB particles, the friction coefficient, the roughness of the surface, and the wear of metal. It shows that the synergistic effect of MWCNT/CF and CB particles is best when the CF content of the rubber matrix is 5 phr, showing the most stable spatial network structure, the best dispersion of CB particles, and minor wear on the end face of the internal mixer.

## 1. Introduction

As an essential piece of mixing equipment, long working hours can cause wear and tear on the end surfaces of an internal mixer. The wear of the end face will increase the gap between the chamber and the end face, which in turn leads to material leakage, reduces the mixing effect, and ultimately affects the rubber performance. Therefore, it is essential to study the end face metal wear during the blending process. This research is of great significance for improving the performance of the internal mixer and reducing metal wear. At the same time, this study also plays an essential role in ensuring the quality of rubber and improving the mixing effect. The rotor and the end face of the internal mixer are shown in Figure 1.

To optimize the wear resistance of rubber and its mechanical properties, researchers have incorporated multi-walled carbon nanotubes (MWCNTs) in the rubber compounding process [1]. Muataz Ali Atieh [2] investigated the effect of MWCNTs on the mechanical properties of natural rubber and found that the initial modulus of rubber with added MWCNTs increased by 12 times compared to pure natural rubber. The unique structure of MWCNTs gives them ultra-high strength, outstanding toughness, and excellent electrical and thermal conductivity, and their use as reinforcing agents can significantly improve the physical properties of composites [3]. It has been reported that Young’s modulus of MWCNTs can be theoretically estimated to be as high as 5 TPa. The experimentally measured mean Young’s modulus of MWCNTs is 1.8 TPa, and the bending strength is 14.2 GPa, which is 100 times stronger than steel, while the density is only 1/7 to 1/6 that of steel; it is unchanged in air below 973 K and has good chemical and thermal stability [4,5,6,7]. Since 1994, when Ajayan [8] incorporated carbon nanotubes into a polymer matrix and prepared polymer/carbon nanotube composites, researchers have carried out partial work [9,10] in an attempt to make carbon nanotubes fully untwisted and uniformly dispersed in polymer matrices by nanocomposite technology to prepare new high-performance multifunctional polymer-based nanocomposites. The SEM images of MWCNTs and CFs are shown in Figure 2.

Carbon fibers (CFs), obtained by pre-oxidation and carbonization of the original filament, have a carbon content of more than 90% and can be compounded with plastics, rubber, metals, and other carbon materials while maintaining the universal properties of carbon materials, and are generally used as reinforcements for composite materials [11,12,13]. At the atomic level, carbon fibers are similar to graphite in that they are composed of layers of carbon atoms arranged in a hexagonal pattern (graphene sheets). The difference between the two lies in the way the layers are connected. Graphite is a crystalline structure with loose interlayer connections, while carbon fibers are not crystalline, and the interlayer connections are irregular. This prevents slippage and enhances the strength of the material [14,15,16]. Yunping Jiang [17] investigated the effect of the blending process on the properties of short-cut carbon fiber/rubber composites. The study showed that the hardness of rubber composites was increased. In addition, the dispersion of short-cut carbon fibers in the composites was highly dependent on the amount of short-cut carbon fibers added, and the rubber composites showed a good distribution when the number of carbon fibers added was 5 phr. Chen Chuangfa [18] investigated the effect of the CF/MWCNT ratio on rubber properties. It was found that the addition of CFs led to better properties when the MWCNTs content was 4 phr.

The literature shows that when mixing CFs and MWCNTs as fillers in rubber composites, the two exhibit synergistic effects in the spatial structure [19,20,21,22,23,24,25]. In this paper, MWCNT/CF/CB composites were prepared by mechanical blending using CB, CFs, and MWCNTs as fillers. Based on the studies of domestic and foreign scholars, this paper investigates the effect of adding different amounts of CFs to the rubber matrix at an MWCNT content of 4 phr on the frictional wear of metal on the end face of the internal mixer.

## 2. Experiment

### 2.1. Main Instruments and Equipment

#### 2.1.1. Internal Mixer

Internal mixer, XSM-500, Shanghai Kechuang Rubber and Plastic Machinery Co., Ltd. (Shanghai, China).

#### 2.1.2. Open Mixer

Open mixer, BL-6157, Baolun Precision Testing Instrument Co., Ltd. (Dongguan, China).

#### 2.1.3. Flat Vulcanizer

Flat Vulcanizer, QLB-400X400X2, Shanghai No. 1 Rubber Machinery Factory (Shanghai, China).

#### 2.1.4. CSM Friction and Wear Testing Machine

CSM Friction and Wear Testing Machine, Tribometer, Peseux, Switzerland.

#### 2.1.5. 3D Laser Measuring Microscope

3D Laser Measuring Microscope, LEXT OLS5000, Olympus, Tokyo, Japan.

#### 2.1.6. RPA2000 Rubber Processing Performance Analyzer

RPA2000 Rubber Processing Performance Analyzer, Alpha, Akron, OH, USA.

#### 2.1.7. Carbon Black Dispersion Meter

Carbon Black Dispersion Meter, DisperGRADER, Alpha, Akron, OH, USA.

#### 2.1.8. Field Emission Scanning Electron Microscope (SEM)

Field Emission Scanning Electron Microscope (SEM), Model SU8000, Hitachi Group (Tokyo, Japan).

### 2.2. Materials and Formulation

The following materials were used in this study: 99% multi-walled carbon nanotubes (MWCNTs), inner diameter 3–5 nm, outer diameter 8–15 nm, specific surface area ≥ 250 m^2^/g, Shenzhen Goshen Pilot Technology Co; carbon fibers (CFs), density 1.75 g/cm, monofilament diameter 7 μm, carbon content ≥ 95%, 1500 mesh, Toray Carbon Fiber (Guangdong) Co.; carbon black (CB), N330, ash ≤ 0.5%, particle size 40 nm, Shijiazhuang Hongda Mineral Products Processing Co. Butadiene rubber 9000 (BR9000), Natural Rubber (TSR20), antioxidant 4020, zinc oxide (ZnO), stearic acid desaturase (SAD), accelerator CZ, and sulfur (S) were all industrial-grade commercially available products. The formulations used in this experiment are shown in Table 1.

### 2.3. Specimen Preparation

The rubber was cut into small pieces for easy feeding. The internal mixer speed was set to 70 r/min, and the initial temperature was 40 °C. Firstly, the pure rubbers were blocked to mix evenly. Then, the ingredients were added at 40 s, and the N330 was added twice at 90 s and 140 s, after which the compounds were discharged at 300 s. Finally, the compounds were prepared into 10 mm sheets after being masticated on a two-roll laboratory mill 10 times [26,27,28].

### 2.4. Performance Test

#### 2.4.1. Rubber Processing Performance

The rubber processing properties were tested on RPA 2000, and the RPA 2000 test method was set to silica multi-luthier stable temperature reduction. The silica Durocher Stabilized Cool Down scan test conditions were 0.01 Hz scan frequency, 0.28–40% scan strain range, and 60 °C temperature. The dynamic modulus G′ curve with strain was obtained.

Payne effect: storage and loss modulus versus strain amplitude at T = 60 °C. The Payne effect is related to the distribution of the internal crosslinking network of the rubber. Generally, the more pronounced the Payne effect is, the more dense the internal crosslinking network of the rubber and the less dispersible the filler [29,30,31].

#### 2.4.2. CSM Friction Wear Test

CSM molds were used to cut the blended rubber specimens with flat surfaces to obtain round rubber specimens of 100 mm diameter. After calibrating the CSM friction and wear test machine, the parameters of the CSM were set to the mixing process parameters. The pressure was set to 5 N, the speed to 70 r/min, and the friction time to 60 min for better observation of the wear effect. To accurately simulate the mixing conditions of the compactor, the material of the metal grinding head used for CSM testing was identical to the end face of the internal mixer. The wear of the metal is most intense during the high-temperature phase of the mixing process. The higher the temperature, the more intense the movement of metal atoms and the easier the wear. To better observe the wear effect, the temperature of the CSM was set to 150 °C. Setting the frequency of CSM to 1 Hz to obtain the friction coefficient allows the instantaneous friction velocity during the mixing process and improves the study’s accuracy [32,33,34,35,36,37,38].

#### 2.4.3. Three-Dimensional Morphological Observation

Analyzing the surface of the test specimen after scanning it with the LEXT OLS5000 Olympus 3D scanner makes it possible to obtain the specimen’s morphological changes and volume changes before and after wear.

#### 2.4.4. Dispersion Test

A rubber section was cut out with a cutter and then tested with a DisperGRADER dispersion meter to obtain dispersion values directly according to ASTM D7723 [39,40,41,42,43,44,45].

#### 2.4.5. SEM

The rubber samples were impregnated in liquid nitrogen, removed, and subjected to brittle fractures to obtain the sections. The sections were characterized using a SU8000 field emission scanning electron microscope.

## 3. Mechanistic Analysis

### 3.1. MWCNT/CF Synergistic Mechanism

The morphology of the filler in the rubber is shown in Figure 3. Figure 3A shows the filler dispersion when filled with 4 phr MWCNT. MWCNTs have a three-dimensional mesh structure with high surface free energy and large surface area, and their surfaces are capable of adsorbing CB particles. Therefore, there are more MWCNT/CB aggregates in the rubber at this time.

Figure 3B shows the filler dispersion when a small amount of CFs are added. There is a synergistic interaction between MWCNT/CB particles and CFs within the rubber matrix, and MWCNT/CB particles attach to CFs. Due to the small amount of CFs, MWCNT/CB particles could not be fully adsorbed onto the CFs. CB particles have the property of being highly agglomerated, so there are many CB particle aggregates in the rubber matrix at this time. Due to the small amount of CF added, the spatial network structure was not formed within the rubber matrix at this time.

Figure 3C shows the dispersion of the filler when the appropriate amount of CFs is added. With the increase in CF incorporation, the total surface area of CFs within the rubber matrix increased, and the number of CB particles and MWCNTs that CF could adsorb onto increased. When the amount of CFs added was 5 phr, the CB particles and MWCNTs inside the rubber were completely adsorbed onto the CF. The CFs within the rubber matrix are also synergistic with each other, and at this time, the filler morphology within the rubber presents a perfect state. A complete spatial network structure is formed within the rubber matrix.

As the amount of CFs added continues to increase, the interaction force between CFs leads to its poor bonding ability with rubber molecules, making CFs appear in the rubber matrix as a serious agglomeration phenomenon. Along with the accumulation between CFs, severe agglomeration between MWCNT/CB particles was also observed. At the same time, the agglomerates of CFs also hinder the dispersion of CB particles, making the rubber’s internal packing state disorderly. The state at this point is shown in Figure 3D.

Figure 3E shows the filler dispersion without the addition of CF/MWCNT. The principal filler in the rubber matrix is CB particles, but since CB particles are highly agglomerated, CB particles form large agglomerates in the rubber matrix.

### 3.2. Metal Wear Mechanism

The frictional wear of rubber on metal in this study is discussed in three main aspects.

(1)Carbon black N330 is the most widely used high abrasion-resistant carbon black. N330 can give the rubber grain better tensile properties, anti-tear properties, abrasion resistance, and elasticity. Due to the quality of raw carbon black oil and the production process control, some aromatic oils are adsorbed onto the surface of carbon black products. These aromatic oils form a thin film on the rubber surface. The oil film reduces the coefficient of friction and reduces friction during the mixing process, thus reducing wear.(2)MWCNTs have good strength, elasticity, fatigue resistance, and isotropy. The mesh structure of MWCNTs can also reduce the wear of metal by isolating the direct contact between rubber and metal.(3)Carbon fibers are lighter in mass than metallic aluminum but more robust than steel. They are lightweight and have high hardness, high strength, high chemical resistance, and high-temperature resistance. Carbon fibers have the inherent, intrinsic characteristics of carbon materials but also the softness and processability of textile fibers, a new generation of reinforcing fibers. The spatial structure of carbon fibers also reduces wear on the metal by isolating the rubber from direct contact with the metal.

## 4. Experimental Results

### 4.1. SEM

The SEM photographs of the rubber specimens are shown in Figure 4. Figure 4(C1) shows more MWCNT/CB agglomerates in the rubber matrix. MWCNTs have a three-dimensional mesh structure, high surface free energy, large surface area, and the ability to adsorb CB particles onto their surface. Therefore, there are more MWCNT/CB agglomerates in the rubber.

Figure 4(C2,C3) show that the amount of CF in the rubber matrix gradually increases, that there is a synergistic effect between MWCNT/CB and CFs in the rubber matrix, and that MWCNT/CB particles are attached to CFs. Due to the small amount of CFs, MWCNT/CB particles could not be fully adsorbed onto CFs. CB particles are easily agglomerated, so there are many CB particle agglomerates in the rubber matrix at this time. Due to the small amount of CFs added, the spatial network structure was not formed within the rubber matrix.

The packing network formed by CF/MWCNT synergism can be seen in Figure 4(C4-1). Figure 4(C4-2) shows MWCNT/CB synergy on the surface of CFs. CB particles and MWCNTs are adsorbed onto the surface of CFs, reducing the accumulation of CB particles and MWCNTs, and CFs have a bridging role in the whole filler network. The CFs within the rubber matrix also synergize, and the filler morphology within the rubber presents a perfect state at this time. A complete spatial network structure is formed within the rubber matrix. This is consistent with the analysis of the packing network above, and the accuracy of the theoretical analysis in Section 3.1 above is verified.

It can be seen from Figure 4(C5,C6) that the number of agglomerates in the rubber matrix gradually increases with the increase of CF content. This is because the interaction force between CFs leads to poor bonding ability between CFs and rubber chains. Moreover, along with the aggregation between CFs, the mixtures of MWCNT/CB occur in severe agglomeration at once. Even more interesting is that the rubber fillers are disordered because the agglomerates of CFs restrict the dispersion of CB. At this time, the CF content has exceeded the optimal synergistic range of MWCNTs (4 phr), which is not conducive to the total dispersion of fillers.

The fillers of the rubber matrix are CB particles when the mixtures of CF/MWCNT are not added. As shown in the Figure 4(C7), the CB particles could form larger agglomerates in the rubber matrix since they are very easily agglomerate.

### 4.2. Rubber Processing Performance Analysis

The processing performance of rubber is shown in Figure 5.

Figure 5A shows the variation of the dynamic modulus G′ with strain, and Figure 5B shows the Payne effect of the rubber. C7 is the rubber without MWCNT/CF addition, and it can be seen that the dynamic modulus of the rubber was significantly increased after the mechanical blending of MWCNTs with CFs. The reasons for this are mainly as follows. On the one hand, the number of filler particles within the rubber matrix is proportional to the dynamic modulus. Adding MWCNT/CF into the rubber matrix increases the internal filler particles within the rubber matrix, which increases G′. On the other hand, both MWCNTs and CF have vigorous surface activity, and both MWCNTs and CFs can adsorb CB particles to produce synergistic effects. This causes the filler network inside the rubber matrix and the dynamic modulus G′ to gradually increase.

The Payne effect of the rubber gradually increases as the CF content increases from 0 phr to 9 phr. The reasons for this are as follows.

(1)The unique spatial network structure of MWCNTs allows them to adsorb a large number of CB particles, which promotes the dispersion of CB particles and reduces the accumulation of CB particles, which helps to reduce the Payne effect. The CF surface is pure and has a large specific surface area. The surface can adsorb CB particles, which promotes the dispersion of CB particles and reduces the accumulation of CB particles, which helps to reduce the Payne effect.(2)MWCNTs and CFs will bind more CB particles when they synergize with CB particles, forming a more severe agglomeration phenomenon inside the spatial network structure, increasing the Payne effect.(3)MWCNTs and CFs have very high surface energy and are very prone to self-agglomeration, and inevitably, MWCNTs and CFs will also agglomerate during the mixing process. The agglomerates of MWCNTs and CFs reduce the adsorption of CB particles, and MWCNT agglomerates make the filler dispersion weaker. Thus, the Payne effect increases. Combined, the effect of the first effect is relatively minor, so the Payne effect as a whole shows an increasing trend, and the filler dispersion decreases.

When the CF content in the rubber matrix is 5 phr, the spatial network structure formed by MWCNTs and CFs synergistically is more mature, and the Payne effect of the rubber tends to be stable. When more CF is added, the filler network inside the rubber matrix does not increase much, so the Payne effect rises slowly.

### 4.3. Filler Dispersibility

The dispersion images are shown in Figure 6, and the dispersion values are shown in Table 2. The dispersion test shows the dispersion of CB particles in the rubber matrix, which in turn gives the overall dispersion status of the filler. The dispersion results are consistent with the Payne effect, and conclusions can be drawn. The filler dispersibility gradually decreased with the increase of CF content when the MWCNT content was 4 phr. When the CF content exceeded 5 phr, CFs showed severe agglomeration in the rubber matrix. At the same time, the agglomerates of CFs also hindered the dispersion of CB particles, which decreased the filler dispersion inside the rubber.

### 4.4. CSM Friction Wear Experiments

#### 4.4.1. Friction Coefficient

Figure 7 shows the friction coefficient curve. The fillers of the C1 formulation are mainly CB particles and MWCNTs. CB particles are readily adsorbed with MWCNTs to produce agglomeration, while CB particles also have a solid self-agglomeration ability. Therefore, there are more CB particle agglomerates and CB/MWCNT agglomerates within the rubber matrix, and these agglomerates make the friction coefficient larger. However, as the blending proceeds, the aromatic oil in the CB particles spills out onto the rubber surface and forms an oil film. The presence of the oil film reduces fluctuations in the mixing process and makes the friction process smoother. Therefore, the coefficient of friction of C1 formulation rubber is higher but shows less fluctuation.

When the CF content was 1 phr, CB particles and MWCNTs were adsorbed onto CFs, which promoted the dispersion of CB particles and reduced the accumulation of CB particles. However, only some of the CB particles and MWCNTs could be adsorbed onto the CF surface due to the small amount of CFs added. At the same time, CB has the accumulation property, so there are many CB particle agglomerates inside the rubber. The CB particle agglomerates make the surface of the compound uneven, so the friction coefficient fluctuates more but decreases relative to C1. CFs hinder the spillage of aromatic oil from CB particles, and the oil film formed on the rubber surface is incomplete. Overall, the friction coefficient is reduced.

When the content of CF was 3 phr, a spatial network prototype was formed between CFs within the rubber matrix, and more CB particles and MWCNTs were adsorbed onto the CF surface. The dispersion of CB particles and MWCNTs further increased, and the particle agglomerates inside the rubber matrix decreased. This also increases the rubber surface’s flatness and reduces the friction coefficient. Meanwhile, with the increase in CF content, the spillage of aromatic oil in CB particles decreases, and the oil film formed on the rubber surface is further reduced, which increases the friction coefficient. The oil film has less influence on the friction coefficient, so the friction coefficient still decreases.

At a CF content of 5 phr, the CFs became entangled with each other, and the CB particles and MWCNTs inside the rubber synergized perfectly with the CFs. CF/MWCNT particles form a well-established spatial network structure within the rubber matrix. On the one hand, the spatial network structure formed by the synergistic CF/MWCNT/CB promotes the dispersion of CB particles. It reduces the CB particle agglomerates within the rubber matrix. On the other hand, the spatial network structure hinders the direct contact between rubber and metal, making the friction coefficient lower. The increase in CF content makes the spatial network structure an enhanced obstacle to the spillover of aromatic oils from CB particles, which further reduces the formation of oil films on the surface of the blends.

As the CF content continues to increase, CFs become entangled with each other, CFs show a strong interaction force, and CFs show a serious agglomeration phenomenon in the rubber matrix. The agglomerates of CF hinder the dispersion of CB particles, and the agglomerates of CB particles inside the rubber matrix keep increasing. Meanwhile, the agglomerates of CF seriously hinder the spillage of aromatic oil from CB particles. This causes the friction coefficient curve to fluctuate more and the friction coefficient to rise.

#### 4.4.2. Observation of Metal Grinding Head Morphology

The surface morphology of the metal grinding head before and after friction is shown in Figure 8.

Figure 8 shows the surface morphology of the metal before and after friction, and Figure 9 shows the three-dimensional morphology of the metal surface before and after friction. As seen from Figure 8(C1), after the rubber without CFs rubbed against the metal, many scratches appeared on the metal surface, with no significant change in the craters. As seen from Figure 9(C1), the amount of height change in group C1 before and after friction is 30 μm. The color of the three-dimensional morphology of the metal surface changes significantly before and after the friction, and the overall change is significant, which indicates that the wear is more serious. There are more CB particle agglomerates in the rubber matrix without CFs, and MWCNT/CB particles adsorb into larger agglomerates with each other, which causes severe metal wear.

Figure 8(C5,C6) and Figure 9(C5,C6) show that the scratches on the metal surface gradually decrease, and the change of craters on the metal surface gradually decreases after the rubber with 1 phr CFs and 3 phr CFs is added to rub against the metal. The height change of group C2 before and after friction was 22.1 μm, and the height change of group C3 before and after friction was 14.9 μm. This indicates that the metal wear decreases as the CF content increases. With the increase in CF content, a synergistic interaction between MWCNT/CB and CFs within the rubber matrix occurs, and MWCNTs and CB particles attach to CFs. The particle agglomerates within the rubber matrix decrease, reducing the metal wear.

As can be seen from Figure 8(C4) and Figure 9(C4), when the rubber with 5 phr CFs added to the metal rubbed against the metal, the metal surface scratches were significantly reduced, and there was no significant change in the metal surface craters. The height change in the C4 group before and after the friction is 6.5 μm. This indicates that the wear of the metal is lower when the CF content in the rubber is 5 phr. When the CF content was 5 phr, the CB particles and MWCNTs inside the rubber were completely adsorbed onto CFs. The CFs within the rubber matrix are also synergistic with each other, and at this time, the filler form within the rubber presents a perfect state. A complete spatial network structure is formed within the rubber matrix. On the one hand, the spatial network structure formed by the synergistic CF/MWCNT/CB promotes the dispersion of CB particles. It reduces the CB particle agglomerates within the rubber matrix. On the other hand, the spatial network structure hinders the direct contact between rubber and metal, resulting in less metal wear.

Figure 8(C5,C6) and Figure 9(C5,C6) show that the height change of group C5 before and after friction is 19.3 μm, and the height change of group C6 before and after friction is 28.9 μm. This indicates that the metal’s wear increases when the rubber’s CF content exceeds 5 phr. CFs have considerable free energy on their surface, and the strong interaction force between CFs leads to their poor bonding ability with rubber molecules, which makes CFs appear in the rubber matrix with a serious agglomeration phenomenon. The aggregation between CFs and severe agglomeration between MWCNT/CB particles was also observed. At the same time, the agglomerates of CFs also hinder the dispersion of CB particles, which leaves the internal packing state of the rubber in a disordered state. The disordered spatial network structure hinders the dispersion of CB particles and increases the CB particle agglomerates within the rubber matrix, which increases metal wear.

From Figure 8(C7) and Figure 9(C7), it can be seen that the amount of height change of the C5 group before and after friction is 36 μm. This indicates that the rubber without adding CF/MWCNT wears the metal more when rubbed against it. This is because CB particles have the accumulation property, and CB particles will form more and larger agglomerates in the rubber matrix. This leads to large fluctuations in the friction process and severe wear.

As can be seen from Figure 10(C1), after the rubber without CF rubbed against the metal, many height profile spikes were ground down, while some pits appeared in the height profile, indicating more severe wear. Figure 9(C2,C3) show the part of the height profile spikes being smoothed out, and the number of craters in the height profile gradually decreases after the rubber with 1 phr CFs and 3 phr CFs is rubbed against the metal. This indicates a continuous reduction in wear and tear. As can be seen from Figure 9(C4), after the rubber with 5 phr CFs was added and rubbed against the metal, some of the height profile spikes were smoothed out, and only a few pits appeared in the height profile, which indicates lighter wear. As can be seen from Figure 9(C5–C7), the metal height profile changes significantly before and after the friction, and the wear is more serious.

#### 4.4.3. Metal Surface Roughness (Ra)

Figure 11 shows the surface roughness of the metal.

As can be seen from Figure 11, the metal surface roughness increased after friction. The change in metal surface roughness is mainly related to the number of particle agglomerates within the rubber matrix and the oil film, and the presence of the oil film can effectively reduce the amount of metal surface roughness change.

There is no obstruction of aromatic oil in CB particles by CFs in CF-free rubber, and the fragrant oil spills out of the rubber surface to form a complete oil film. The presence of the oil film effectively reduces the amount of roughness variation on the metal surface. However, the lack of adsorption of CFs onto CB particles resulted in the presence of more CB particle agglomerates within the rubber matrix. The CB particle agglomerates make the rubber surface uneven, which causes severe fluctuations during the friction process. Therefore, despite the presence of an oil film, the amount of change in metal surface roughness after rubbing the blended rubber without CF addition with metal is still significant.

With the increase of CF content within the rubber matrix, synergistic interaction between CFs, CB particles, and MWCNTs occurs, and more and more CB particles and MWCNTs are adsorbed on the CF surface. This promotes the dispersion of CB particles and MWCNTs and helps to reduce the number of CB particle agglomerates within the rubber matrix. When the CF content is 5 phr, the CFs entangle, the CB particles and MWCNTs inside the rubber synergize perfectly with the CFs, and the vast majority of CB particles and MWCNTs are adsorbed onto the CF surface. There are very few CB particle agglomerates at this time inside the rubber, and the filler form inside the rubber is a perfect spatial network structure. The spatial network structure formed by the synergistic CF/MWCNT/CB particles hinders the direct contact between rubber and metal, all of which flattens the rubber surface and helps reduce the variation in metal surface roughness. At the same time, a small amount of CFs prevents some of the aromatic oils from spilling out, but there is still a more significant amount of aromatic oils spilling out to form an oil film. This also reduces the amount of change in metal surface roughness.

When the CF content exceeds 5 phr, severe CF agglomeration occurs within the rubber matrix. The agglomerates of CFs also hinder the dispersion of CB particles and MWCNTs, increasing particle agglomerates within the rubber matrix and decreasing the flatness of the rubber surface. At the same time, the agglomerates of CFs seriously hinder the spillage of aromatic oil from CB particles and prevent the formation of the oil film. This all makes the amount of variation of metal surface roughness increase.

## 5. Conclusions

In this experiment, the frictional wear of rubber and metal during the blending process was analyzed by studying the formation mechanism of the CF/MWCNT/CB synergistic network. The addition of CFs resulted in a synergistic effect with MWCNTs and CB in the rubber matrix and improved the dispersion of CB particles. However, as the CF content within the rubber matrix increases, the CF itself produces agglomerates. The agglomerates of CFs make the spatial meshwork of the rubber stack and become disordered, resulting in a gradual decrease in the dispersion of CB particles. In addition, the spatial meshwork co-constructed by CF/MWCNT/CB could restrict the overflow of aromatic oil from the rubber surface, so that the wear between the rubber and metal and the change of metal roughness gradually decrease over the whole mixing process. When the amount of CFs added is 5 phr, the CFs in the rubber matrix entangle, CB particles and MWCNTs synergize perfectly with CF, and most of the CB particles are adsorbed onto the CF surface. There are very few CB particle agglomerates at this time inside the rubber, and the filler form inside the rubber is a perfect spatial network structure. The metal wear is reduced to the minimum at this time. When the CF content exceeds 5 phr, the spatial network structure starts to stack and become disordered, which occurs increasingly more with the increase in CF content, and the amount of metal wear and metal roughness change keeps increasing with it. However, when the amount of CFs is more than 5 phr, the spatial meshwork begins to become stacked and disordered. As the amount of CFs increases, the spatial meshwork is more disordered, which makes the wear of metal and the change of metal roughness increase accordingly.

## Figures and Tables

**Figure 1 polymers-14-03731-f001:**
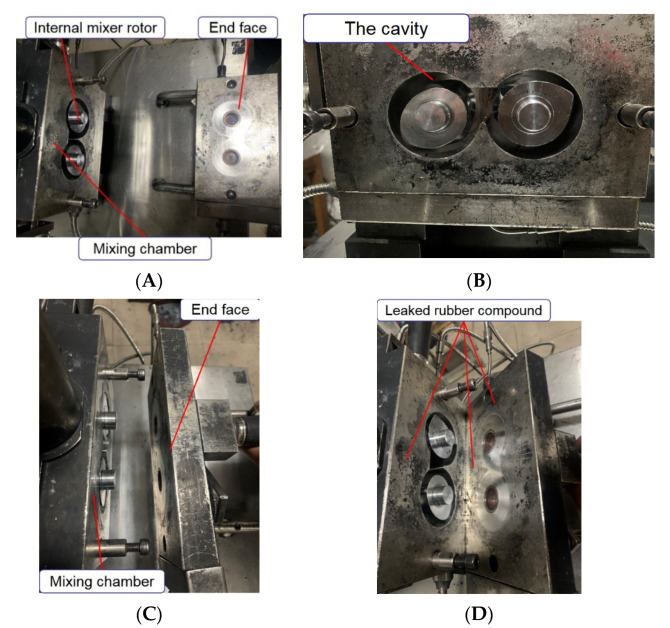
Rotor and end face of the mixer. ((**A**) is a general view, (**B**) is a rotor, (**C**) is a top view, and (**D**) is a side view).

**Figure 2 polymers-14-03731-f002:**
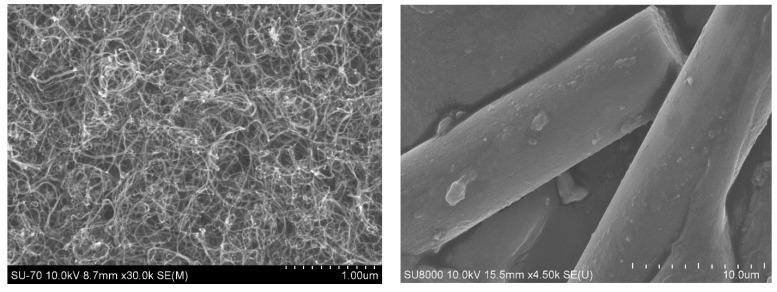
Scanning electron microscopy of MWCNTs (**left**) and CFs (**right**).

**Figure 3 polymers-14-03731-f003:**
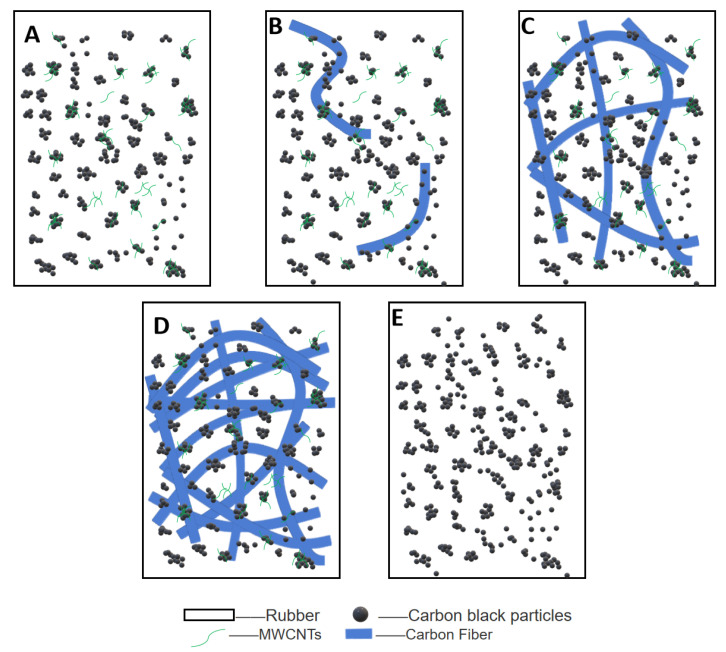
Form of filler in rubber. (**A**) MWCNT/CB; (**B**) A small amount of CF/MWCNT/CB; (**C**) Appropriate amount of CF/MWCNT/CB; (**D**) Excessive CF/MWCNT/CB; (**E**) MWCNT free.

**Figure 4 polymers-14-03731-f004:**
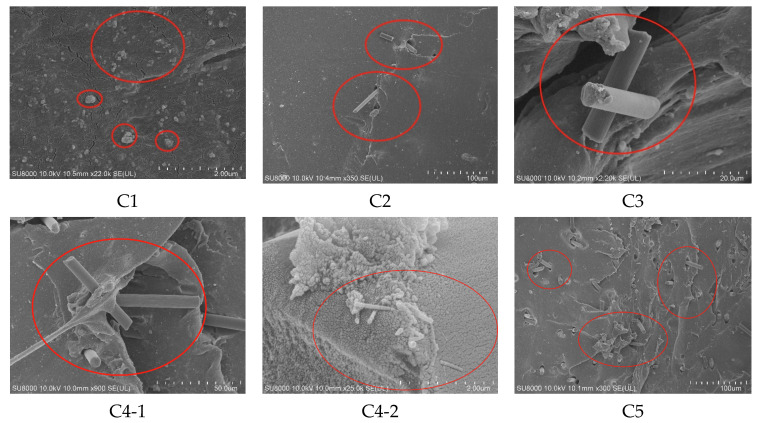
SEM. **C1**–**C7** are 7 groups of rubber.

**Figure 5 polymers-14-03731-f005:**
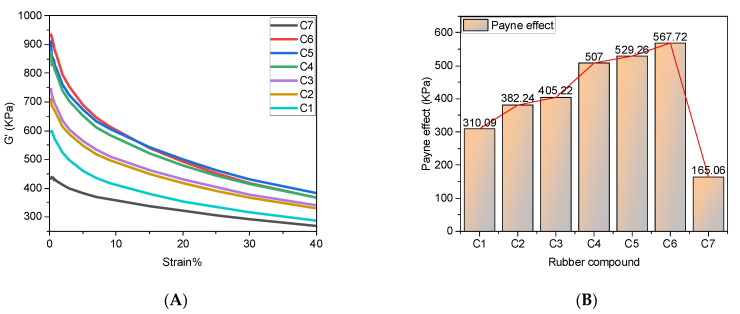
Rubber processing properties. (**A**) Stress-strain curve; (**B**) Payne effect.

**Figure 6 polymers-14-03731-f006:**
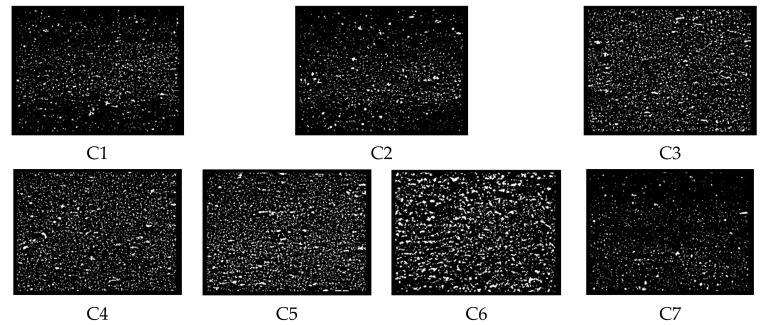
Dispersion images. (**C1**–**C7**) are the dispersion images of 7 groups of rubbers.

**Figure 7 polymers-14-03731-f007:**
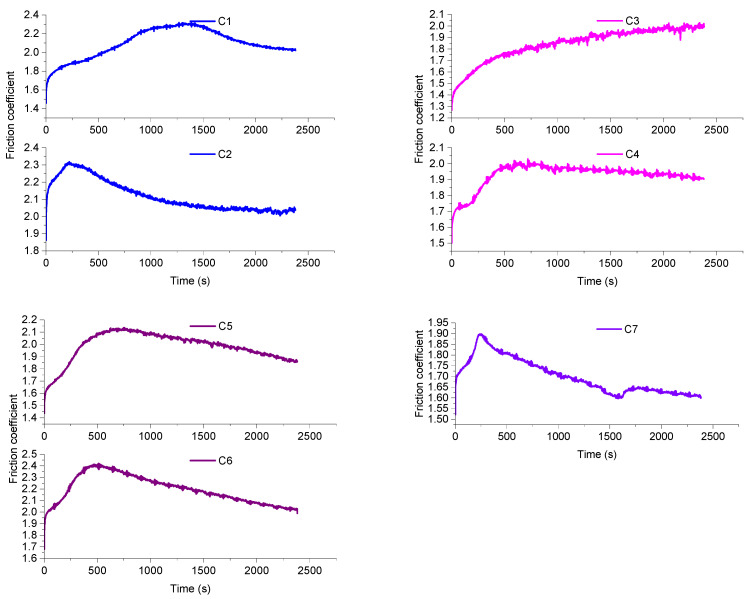
Friction coefficient.

**Figure 8 polymers-14-03731-f008:**
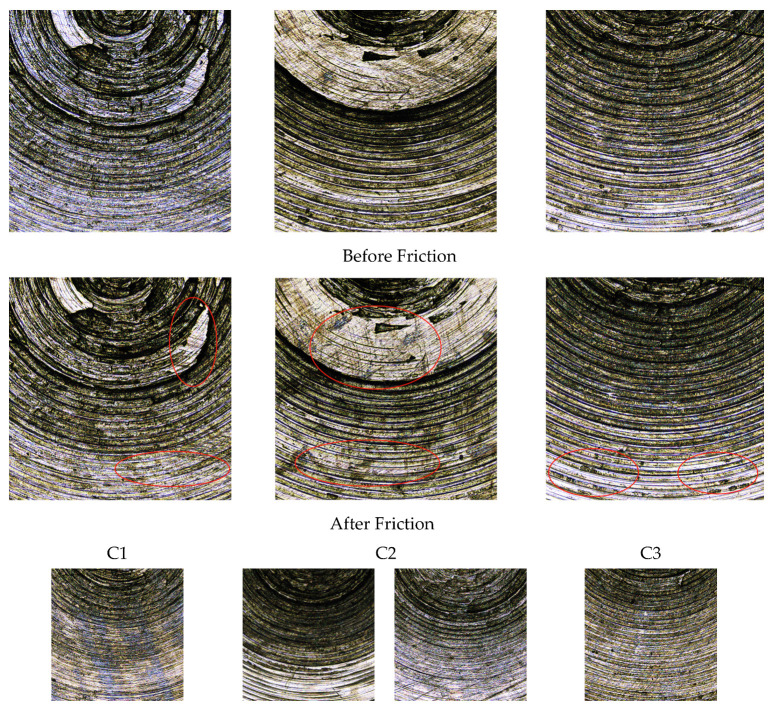
Metal surface morphology. **C1**–**C7** are the surface morphologies of 7 groups of metals before and after friction.

**Figure 9 polymers-14-03731-f009:**
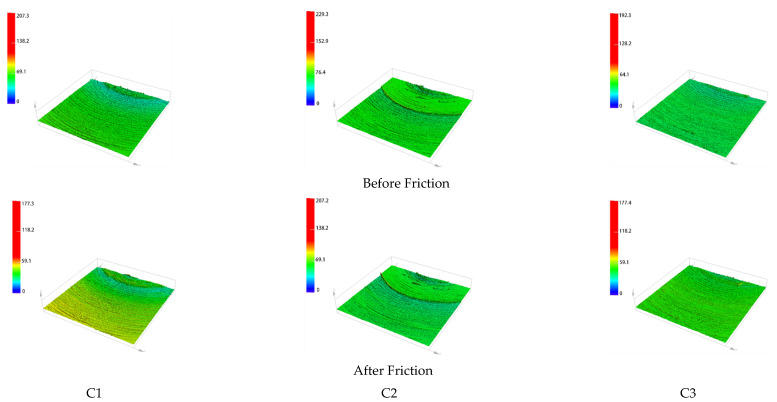
Metal surface 3D morphology. **C1**–**C7** are the three-dimensional surface morphologies of 7 groups of metals before and after friction.

**Figure 10 polymers-14-03731-f010:**
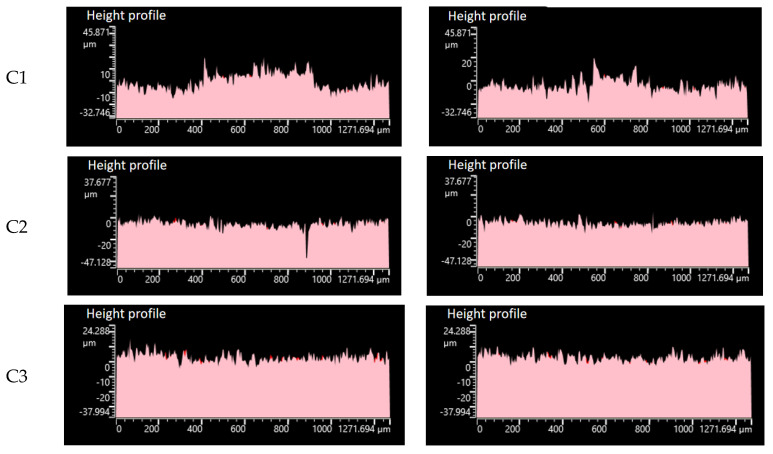
Metal height profile. (**C1**–**C7**) are the height profiles of 7 groups of metal before and after friction.

**Figure 11 polymers-14-03731-f011:**
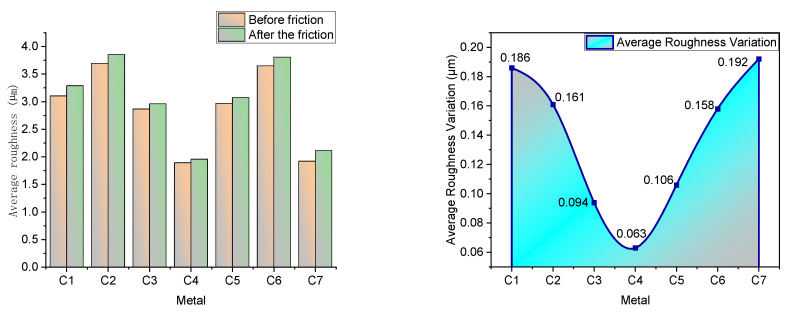
Surface roughness (Ra). The **left** picture shows the roughness before and after friction, and the **right** picture shows the roughness change.

**Table 1 polymers-14-03731-t001:** Formulation.

Raw Material	C1	C2	C3	C4	C5	C6	C7
BR9000	20	20	20	20	20	20	20
TSR20	80	80	80	80	80	80	80
MWCNTs	4	4	4	4	4	4	0
CF	0	1	3	5	7	9	0
N330	30	30	30	30	30	30	30
ZnO	5	5	5	5	5	5	5
SAD	2.5	2.5	2.5	2.5	2.5	2.5	2.5
CZ	2	2	2	2	2	2	2
S	2	2	2	2	2	2	2

**Table 2 polymers-14-03731-t002:** Dispersion values.

Vulcanizate	C1	C2	C3	C4	C5	C6	C7
Dispersion	7.32	7.15	6.26	6.1	5.22	4.45	7.61

## Data Availability

Our team’s data iare accurate and reliable, and the experiments are repeatable. We guarantee that the information is fully available.

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
