# Peer review of "Influence of the Synergistic Effect of Multi-Walled Carbon Nanotubes and Carbon Fibers in the Rubber Matrix on the Friction and Wear of Metals during the Mixing Process"

_polymers, 2022, doi:10.3390/polym14183731_

Round 1

Reviewer 1 Report

This article is devoted to the study of promising rubber-based composites. The presented results are well-written and it is clear that the study was carried out at a high level. The article presents interesting and important results. However, the article requires a minor revision before being published.
1.    MWCNTs in line 14 should be “multiwall carbon nanotubes” (as in line 38)
2.    Carbon black (CB) is not defined in the abstract (line 16)
3.    The scientific novelty of the article should be better described in the introduction section. In the literature, there are a large number of works on MWNT/CF/CB composites in different ratios and for different purposes. A synergistic effect is also described in the literature. Therefore, the uniqueness and significance of the article should be highlighted more clearly.
4.    Line 51 “a lot of work [9-10]” – two articles may be a sign of “a lot of work” if they are review articles, but 9 and 10 are research articles of 4 and 8 pages long correspondingly. Please, rephrase or provide more articles.
5.    Sections 2.1 and 2.2 should be transformed to a text with explanations of each position (as done in section 2.4) instead of the list.
6.    Fig.6 – time axis is given in seconds, however values are only at C7 curve.
7.    Lines 497-498 – the statement should be done by the entire team and not just one person.

Author Response

Dear reviewer#1:

Thank you very much for your valuable comments, which are very important to me. Thank you for your selfless contribution, which greatly improved my article. Sincerely wish you good health and success in your work. Thank you very much!

Question (1): MWCNTs in line 14 should be “multiwall carbon nanotubes” (as in line 38)

Response to comments:

In this paper, multiwall carbon nanotubes (MWCNTs) were blended with carbon fibers (CF) and added to rubber by mechanical compounding to obtain MWCNTs/CF/ carbon black (CB) composites.

Question (2): Carbon black (CB) is not defined in the abstract (line 16)

Response to comments:

In this paper, multiwall carbon nanotubes (MWCNTs) were blended with carbon fibers (CF) and added to rubber by mechanical compounding to obtain MWCNTs/CF/ carbon black (CB) composites.

Question (3): The scientific novelty of the article should be better described in the introduction section. In the literature, there are a large number of works on MWNT/CF/CB composites in different ratios and for different purposes. A synergistic effect is also described in the literature. Therefore, the uniqueness and significance of the article should be highlighted more clearly.

Response to comments:

As the essential mixing equipment, long working hours can cause wear and tear on the end surfaces of the internal mixer. The wear of the end face will increase the gap between the chamber and the end face, leading to material leakage, reducing the mixing effect, and ultimately affecting the rubber performance. Therefore it is essential to study the end face metal wear during the blending process. This research is of great significance for improving the lives of the internal mixer and reducing metal wear. At the same time, this study also plays an essential role in ensuring the quality of rubber and improving the mixing effect.

Question (4): Line 51 “a lot of work [9-10]” – two articles may be a sign of “a lot of work” if they are review articles, but 9 and 10 are research articles of 4 and 8 pages long correspondingly. Please, rephrase or provide more articles.

Response to comments:

Since 1994, when Ajayan [8] incorporated carbon nanotubes as inorganic fillers into polymer matrix and prepared polymer/carbon nanotube composites, researchers have carried did some work [9-10] in an attempt to make carbon nanotubes fully untwisted and uniformly dispersed in polymer matrix by nanocomposite technology to prepare new high-performance multifunctional polymer-based nanocomposites. The SEM images of MWCNTs and CF are shown in Figure 1.

Question (5): Sections 2.1 and 2.2 should be transformed to a text with explanations of each position (as done in section 2.4) instead of the list.

Response to comments:

2.1. Main Instruments and Equipment

2.1.1 Internal mixer

Internal mixer, XSM-500, Shanghai Kechuang Rubber and Plastic Machinery Co., Ltd.;

2.1.2 Open mixer

Open mixer, BL-6157, Baolun Precision Testing Instrument Co., Ltd.;

2.1.3 Flat Vulcanizer

Flat Vulcanizer, QLB-400X400X2, Shanghai No. 1 Rubber Machinery Factory;

2.1.4 CSM-Friction and Wear Testing Machine

CSM-Friction and Wear Testing Machine, Tribometer, Switzerland;

2.1.5 3D Laser Measuring Microscope

3D Laser Measuring Microscope, LEXT OLS5000, Olympus, Japan;

2.1.7 RPA2000 Rubber Processing Performance Analyzer

RPA2000 Rubber Processing Performance Analyzer, Alpha, USA;

2.1.8 Carbon Black Dispersion Meter

Carbon Black Dispersion Meter, DisperGRADER, Alpha, USA Company;

2.1.9 Field Emission Scanning Electron Microscope (SEM)

Field Emission Scanning Electron Microscope (SEM), Model SU8000, Hitachi Group.

Thank you for your comments, but due to the specificity of the experimental material, chapter 2.2 can be done without separation.

Question (6): Fig.6 – time axis is given in seconds, however values are only at C7 curve.

Response to comments:

4.4. CSM Friction Wear Experiments

4.4.1. Friction Coefficient

Figure 6. Friction coefficient.

Question (7): Lines 497-498 – the statement should be done by the entire team and not just one person.

Response to comments:

Our team's data is accurate and reliable, and the experiments are repeatable. We guarantee that the information is fully available.

Reviewer 2 Report

In my opinion the article: "Influence of the synergistic effect of multi-walled carbon nanotubes and carbon fibers in the rubber matrix on the friction and wear of metals during the mixing process", contains quite interesting information and is quite correctly structured from a scientific point of view. The problem is that the English seems to be quite poor and this leads to some difficulties in the understanding of some wide sections of the manuscript, as well as of in the understanding of the authors massage regarding results and the related explanation (discussion). On this regards, the authors are invited to improve this issue by mean of the help of a mother tongue. For this reason, the manuscript, after to be published, has to be widely rewritten, significantly improving the English language. From a scientific base point, no particular critical aspects have been found.

Here some question:

·         Abstract: many times the authors claim: "the end face of the internal mixer". On this regards, the authors are invited to better explain what they mean with "end face". In my opinion, also a scheme of the mentioned apparatus can be presented to facilitate the readers on this aspect.

·         Abstract: why did the authors use the term "Mesh-work" to indicate something which seems to have the features of a "network"?

·         Also in the abstract English language has to be a bit improved;

·         Pag. 2/19 - Raws 49 and 50 -  The authors claim: "Since 1994, when Ajayan [8] incorporated carbon nanotubes as inorganic fillers into polymer matrix and...";

- Why do the authors use "inorganic filler" in the case of carbon nanotubes?

·         Pag. 2/19 - Raws 67 and 78 -  The authors claim: "Studies have shown that short-cut carbon fibers have improved rubber composites' hardness, constant tensile stress, and tensile strength.".

- The authors are invited to explain what they mean when they write that short-cut carbon fibers have improved constant tensile stress. Do they mean that the addition of carbon fibers doesn't lead to any changes in the tensile stress?. In any case, this should conflict with the remaining part of the sentence, in which an improvement of tensile strength is claimed;

·         Pag 3. - Table 1. From the described framework appears as the only variable is the amount of carbon fibers, while no changes in the weight fraction of other fillers has been studied. In my opinion, this procedure is very restrictive from a scientific base point, as it hinders any possibility to optimize the obtained results, through the variation also in the content of the other fillers. In my opinion, as an alternative, a lower amount of percentage of CF could be investigated, together with also 3-5 percentages of MWCNT.

- Could the authors provide some scientifically valid justification regarding their choice?

·         Pag. 3. In my opinion, paragraph 2.3 has to be completely rewritten, eliminating any imperative verbal form;

·         Pag. 4 - Raw 126 and 127. The authors claim: "Some studies have shown that the metal wear is most intense in the high-temperature stage of the mixing process,...";

- The authors are invited to provide literature references regarding the mentioned "studies";

·         Pag. 5. Figure 2. In the related caption any reference to part 2E is completely missing. In my opinion, the addition of some (brief) details is mandatory;

·         Pag. 5. In my opinion, also in the paragraph 3.1 the English language be significantly improved;

·         Pages 6 and 7 - Figure 3. The authors are invited to provide a wide amount of details in the related caption (mandatory);

·         Pag. 7. Raws 218 and 219. The authors claim: "This is consistent with the analysis of the packing network above, and the accuracy of the above theoretical analysis is verified.".

- The authors are invited to explain why the "...above theoretical analysis" is verified. If needed, they can provide some literature references.

·         Pag. 7. Raws 220 to 226. The authors claim: "Figure 3 C5 and C6 show that the number of agglomerates in the rubber matrix gradually increases with the increasing CF content. It indicates that the CF content has exceeded the optimal synergistic content of 4 phr MWCNTs at this time, which is not conducive to adequate filler dispersion. Figure 3 C7 shows the filler dispersion without adding CF/MWCNTs. The principal filler in the rubber matrix is CB particles, but since CB particles are highly agglomerated, CB particles form large agglomerates in the rubber matrix.

- Despite the message appears quite clear and can be agreed, the authors are invited to completely rewrite this sentence, as the related readability is very poor. A significant improvement in the English language is sufficient;

·         Pag. 10. Raw 306. The authors claim: "At a CF content of 5phr, the CFs cross-linked with each other,..";

- Probably the term cross-link isn't completely appropriate in this case, because it is generally referred to tridimensional network rising from mere chemical bonding. In my opinion, the term "entanglement" could be more appropriate. As resulting from the manuscript, the described phenomenon seems to refers to a microstructure in which the CF physical percolation threshold has been overcame. In this framework a continuous path between CF themselves has been formed and they touch each others.

- The authors are invited to provide their point on view on this topic;

·         Figures 10 and 11. The authors are invited to provide more details in the related caption;

·         Pag 16. Raw 461. The authors claim: "When the CF content is 5phr, the CFs lap each other".

- The authors are invited to specify what do they mean with "lap each other";

·         Pag. 16. Raw 469. The authors claim: "...a more significant amount of scented oil spilling...".

- The authors are invited to specify what do they mean with "scented oil";

·         Pag. 17. Raws 482 to 488. The authors claim: "The agglomerates of CF cause the spatial network structure inside the rubber matrix to stack, the spatial structure inside the rubber matrix to be disturbed, and the dispersion of CB particles to gradually decrease. The amount of wear between rubber and metal and the change in metal roughness gradually decreases during the mixing process. ".

- The authors are invited to rewrite the sentence, improving the English language, despite the related message could be understood and agreed;

·         Pag. 17. Raw 494 to 495. The authors claim: "When the content of CF exceeds 5 phr, the spatial network structure starts to stack and disorder and increases with the increase of CF content..."

- Also in this case, the authors are invited to rewrite the sentence, improving the English language.

Author Response

Dear reviewer#2:

Thank you very much for your valuable comments, which are very important to me. Thank you for your selfless contribution, which greatly improved my article. Sincerely wish you good health and success in your work. Thank you very much!

Question (1): Abstract: many times the authors claim: "the end face of the internal mixer". On this regards, the authors are invited to better explain what they mean with "end face". In my opinion, also a scheme of the mentioned apparatus can be presented to facilitate the readers on this aspect.

Response to comments:

Thank you for your comments. I have explained the device in the article.

As the essential mixing equipment, long working hours can cause wear and tear on the end surfaces of the internal mixer. The wear of the end face will increase the gap between the chamber and the end face, which in turn leads to material leakage, reduces the mixing effect, and ultimately affects the rubber performance. Therefore, it is essential to study the end face metal wear during the blending process. This research is of great significance for improving the lives of the internal mixer and reducing metal wear. At the same time, this study also plays an essential role in ensuring the quality of rubber and improving the mixing effect. The rotor and the end face of the internal mixer are shown in Figure 1.

A

B

C

D

Figure 1. Rotor and end face of the mixer

Question (2): Abstract: why did the authors use the term "Mesh-work" to indicate something which seems to have the features of a "network"?·

Response to comments:

This is due to macromolecular chains within the rubber matrix and the formation of a mesh structure between the filler and the macromolecular chains. Therefore, I have applied the term "reticulation" in the text.

Question (3):·Also in the abstract English language has to be a bit improved;

Response to comments:

Thanks for your comments. I have revised the language.

As a high-intensity running equipment, the wear of the internal mixer determines the quality of rubber and the life of them. In general, the wear of the internal mixer is caused by the friction between the rubber and metal during the mixing process, and the most severe wear position is the end face of the equipment. In this paper, the mixture of multiwall carbon nanotubes (MWCNTs) and carbon fibers (CF) are added to rubber by mechanical compounding to obtain MWCNTs/CF/ carbon black (CB) composites. By investigating the synergistic mechanism of MWCNTs and CF, we analyze the effect of the MWCNTs/CF ratio on the frictional wear of metal on the end face of the internal mixer. At the microscopic level, MWCNTs and CF form a spatial meshwork with CB particles through synergistic interactions. On the one hand, the CB particles could be adsorbed on the spatial meshwork to promote the dispersion of CB particles. On the other hand, the formation of oil film could also be slowed down due to the spatial meshwork which could hinder the spillage of aromatic oil. Meanwhile, the spatial meshwork serves as a physical isolation layer between the rubber and metal to reduce friction. Therefore, it dramatically impacts the dispersion degree of CB particle, the friction coefficient, the roughness of the surface, and the wear of metal. It shows that the synergistic effect of MWCNTs/CF and CB particles is the best when the content of CF in the rubber matrix is 5 phr, with the most stable spatial network structure, the best dispersion of CB particles, and minor wear on the end face of the internal mixer.

Question (4):·Pag. 2/19 - Raws 49 and 50 -  The authors claim: "Since 1994, when Ajayan [8] incorporated carbon nanotubes as inorganic fillers into polymer matrix and...";

- Why do the authors use "inorganic filler" in the case of carbon nanotubes?

Response to comments:

Since my unclear expression caused you trouble, I have revised it in the original text. I will explain it in detail to you. The point I was trying to make in the text is that carbon nanotubes are inorganic fillers. This is because carbon nanotubes are composed of inorganic carbon monomers.

Since 1994, when Ajayan [8] incorporated carbon nanotubes into polymer matrix and prepared polymer/carbon nanotube composites, researchers have carried out partial work [9-10] in an attempt to make carbon nanotubes fully untwisted and uniformly dispersed in polymer matrix by nanocomposite technology to prepare new high-performance multifunctional polymer-based nanocomposites.

Question (5):·Pag. 2/19 - Raws 67 and 78 -  The authors claim: "Studies have shown that short-cut carbon fibers have improved rubber composites' hardness, constant tensile stress, and tensile strength.". The authors are invited to explain what they mean when they write that short-cut carbon fibers have improved constant tensile stress. Do they mean that the addition of carbon fibers doesn't lead to any changes in the tensile stress?. In any case, this should conflict with the remaining part of the sentence, in which an improvement of tensile strength is claimed;

Response to comments:

Thank you for your comments, as my unclear presentation caused a misunderstanding. I have made the changes.

Yunping Jiang [17] investigated the effect of the blending process on the properties of short-cut carbon fiber/rubber composites. The study showed that the hardness of rubber composites was increased. And the dispersion of short-cut carbon fibers in the composites was highly dependent on the amount of short-cut carbon fibers added, and the rubber composites with good distribution when the number of carbon fibers added was 5 phr.

Question (6):·Pag 3. - Table 1. From the described framework appears as the only variable is the amount of carbon fibers, while no changes in the weight fraction of other fillers has been studied. In my opinion, this procedure is very restrictive from a scientific base point, as it hinders any possibility to optimize the obtained results, through the variation also in the content of the other fillers. In my opinion, as an alternative, a lower amount of percentage of CF could be investigated, together with also 3-5 percentages of MWCNT.

Response to comments:

Thank you for your valuable comments, which are very critical to me. I will start the rest of the filler in the follow-up study. Thank you very much.

Question (7):·Could the authors provide some scientifically valid justification regarding their choice?

Response to comments:

Chen Chuangfa [18] investigated the effect of CF/MWCNTs ratio on rubber properties. It was found that the addition of CF was able to obtain better properties when the MWCNTs content was 4 phr.

Therefore, we performed experiments based on the addition of 4phr MWCNTs. Mixing CF and MWCNTs as fillers in rubber composites shows a synergistic effect in the spatial structure [19-25]. This paper used CB, CF, and MWCNTs as fillers to prepare MWCNTs/CF/CB composites by mechanical blending. In this paper, the effect of adding different amounts of CF to the rubber matrix on the friction and wear of the metal on the end face of the internal mixer was studied.

Question (8):·Pag. 3. In my opinion, paragraph 2.3 has to be completely rewritten, eliminating any imperative verbal form;

Response to comments:

Thanks for your comments. I have revised the language.

The rubbers were cut into small pieces for easy feeding. The internal mixer speed was set to 70 r/min, and the initial temperature was 40 °C. Firstly, the pure rubbers were blocked to mix evenly. Then, the ingredients were added at 40s, and the N330 was added twice at 90s and 140s, after which the compounds were discharged at 300s. Fi-nally, the compounds were prepared into 10mm sheets after being masticated on a two-roll laboratory mill 10 times.

Question (9):·Pag. 4 - Raw 126 and 127. The authors claim: "Some studies have shown that the metal wear is most intense in the high-temperature stage of the mixing process,..."; The authors are invited to provide literature references regarding the mentioned "studies";

Response to comments:

Thank you for your comments, as my misrepresentation has caused you to misunderstand. I will give you a detailed analysis. The higher the temperature, the more intense the movement of metal atoms, and the easier the wear. Therefore, the wear is the lowest at high temperatures. I have made changes to the original text.

The higher the temperature, the more intense the movement of metal atoms and the easier the wear. Metal wear is most intense at the high-temperature stage of the mixing process. To better observe the wear effect, the temperature of the CSM was set to 150°C.

Question (10):·Pag. 5. Figure 2. In the related caption any reference to part 2E is completely missing. In my opinion, the addition of some (brief) details is mandatory;

Response to comments:

Thank you for your comments. I have made the addition.

Figure 2E shows the filler dispersion without the addition of CF/MWCNTs. The principal filler in the rubber matrix is CB particles, but since CB particles are highly agglomerated, CB particles form large agglomerates in the rubber matrix.

Question (11):·Pag. 5. In my opinion, also in the paragraph 3.1 the English language be significantly improved;

Response to comments:

Thank you for your compliments. I sincerely appreciate your encouragement and support for my research.

Question (12):·Pages 6 and 7 - Figure 3. The authors are invited to provide a wide amount of details in the related caption (mandatory);

Response to comments:

Thank you for your comments. I have made the addition.

C1

C2

C3

C4-1

C4-2

C5

C6-1

C6-2

C7

Figure 3. SEM.

The SEM photographs of the rubber specimens are shown in Figure 3. Figure 3C1 shows more MWCNTs/CB agglomerates in the rubber matrix. MWCNTs have a three-dimensional mesh structure, high surface free energy, large surface area, and the ability to adsorb CB particles on their surface. Therefore, there are more MWCNTs/CB agglomerates in the rubber.

Figures 3C2 and C3 show that the amount of CF in the rubber matrix gradually increases, and there is a synergistic effect between MWCNTs/CB and CF in the rubber matrix, and MWCNTs/CB are attached to CF. Due to the small amount of CF, MWCNTs/CB could not be fully adsorbed on CF. CB particles are easily agglomerated, so there are many CB particle agglomerates in the rubber matrix at this time. Due to the small amount of CF addition, the spatial network structure has not been formed within the rubber matrix.

The packing network formed by CF/MWCNTs synergism can be seen in Figure 3 C4-1. Figure 3 C4-2 shows MWCNTs/CB synergy on the surface of CF. CB particles and MWCNTs are adsorbed on the CF surface, reducing the accumulation of CB particles and MWCNTs, and CF has a bridging role in the whole filler network. The CF within the rubber matrix also synergizes, and the filler morphology within the rubber presents a perfect state at this time. A complete spatial network structure is formed within the rubber matrix. This is consistent with the analysis of the filler network above, and the accuracy of the above theoretical analysis is verified.

It can be seen from Figure 3 C5 and C6 that the number of agglomerates in the rubber matrix gradually increases with the increase of CF content. This is because the interac-tion force between CF particles leads to poor bonding ability between CF and rubber chains. Moreover, along with the aggregation between CF particles, the mixtures of MWCNTs/CB occur in severe agglomeration at once. Even more interesting is that the rubber fillers are disordered because the agglomerates of CF particles would restrict the dispersion of CB. At this time, the content of CF particles has exceeded the optimal synergistic range of MWCNTs (4phr), which is not conducive to the total dispersion of fillers.

The fillers of the rubber matrix are CB particles when the mixtures of CF/MWCNTs are not added. As shown in the Figure3 C7, the CB particles could form larger agglomerates in the rubber matrix since they are very easy to agglomerate.

Question (13):·Pag. 7. Raws 218 and 219. The authors claim: "This is consistent with the analysis of the packing network above, and the accuracy of the above theoretical analysis is verified.".

- The authors are invited to explain why the "...above theoretical analysis" is verified. If needed, they can provide some literature references.

Response to comments:

Thank you again for your careful and careful review; I will explain it to you in detail. I want to express that this is consistent with the analysis of the dispersion mechanism in 3.1 above. I have edited this sentence in the original text.

The packing network formed by CF/MWCNTs synergism can be seen in Figure 3 C4-1. Figure 3 C4-2 shows MWCNTs/CB synergy on the surface of CF. CB particles and MWCNTs are adsorbed on the CF surface, reducing the accumulation of CB particles and MWCNTs, and CF has a bridging role in the whole filler network. The CF within the rubber matrix also synergizes, and the filler morphology within the rubber presents a perfect state at this time. A complete spatial network structure is formed within the rubber matrix. This is consistent with the analysis of the packing network above, and the accuracy of the theoretical analysis in Section 3.1 above is verified.

Question (14): Pag. 7. Raws 220 to 226. The authors claim: "Figure 3 C5 and C6 show that the number of agglomerates in the rubber matrix gradually increases with the increasing CF content. It indicates that the CF content has exceeded the optimal synergistic content of 4 phr MWCNTs at this time, which is not conducive to adequate filler dispersion. Figure 3 C7 shows the filler dispersion without adding CF/MWCNTs. The principal filler in the rubber matrix is CB particles, but since CB particles are highly agglomerated, CB particles form large agglomerates in the rubber matrix.

- Despite the message appears quite clear and can be agreed, the authors are invited to completely rewrite this sentence, as the related readability is very poor. A significant improvement in the English language is sufficient;

Response to comments:

It can be seen from Figure 3 C5 and C6 that the number of agglomerates in the rubber matrix gradually increases with the increase of CF content. This is because the interac-tion force between CF particles leads to poor bonding ability between CF and rubber chains. Moreover, along with the aggregation between CF particles, the mixtures of MWCNTs/CB occur in severe agglomeration at once. Even more interesting is that the rubber fillers are disordered because the agglomerates of CF particles would restrict the dispersion of CB. At this time, the content of CF particles has exceeded the optimal synergistic range of MWCNTs (4phr), which is not conducive to the total dispersion of fillers.

The fillers of the rubber matrix are CB particles when the mixtures of CF/MWCNTs are not added. As shown in the Figure3 C7, the CB particles could form larger agglomerates in the rubber matrix since they are very easy to agglomerate.

Question (15):·Pag. 10. Raw 306. The authors claim: "At a CF content of 5phr, the CFs cross-linked with each other,..";

- Probably the term cross-link isn't completely appropriate in this case, because it is generally referred to tridimensional network rising from mere chemical bonding. In my opinion, the term "entanglement" could be more appropriate. As resulting from the manuscript, the described phenomenon seems to refers to a microstructure in which the CF physical percolation threshold has been overcame. In this framework a continuous path between CF themselves has been formed and they touch each others.

- The authors are invited to provide their point on view on this topic;

Response to comments:

Thanks for your opinion. I think you are correct. I have edited the original text.

At a CF content of 5phr, the CFs entanglement with each other, and the CB particles and MWCNTs inside the rubber synergized perfectly with the CFs. CF/MWCNTs form a well-established spatial network structure within the rubber matrix. On the one hand, the spatial network structure formed by the synergistic CF/MWCNTs/CB promotes the dispersion of CB particles.

Question (16):·Figures 10 and 11. The authors are invited to provide more details in the related caption;

Response to comments:

Thanks for your input. I've added the clarification and rewritten it.

Figure 10 A shows the volume of the metal surface before and after friction, and Figure 10 B shows the amount of wear. With the increase of CF addition, the wear of metal shows a trend of decreasing and then increasing, which is related to the unique spatial structure of CF. When the content of CF does not exceed 5phr, the CF within the rubber matrix generates a stable spatial network structure with MWCNTs and CB particles in the matrix through synergistic action, giving full play to the excellent performance of the filler. The CB particles and MWCNTs within the rubber matrix are adsorbed on the CF surface, which reduces the accumulation of CB particles and makes the tiny rubber surface flat. Meanwhile, the spatial structure formed by the synergistic CF/MWCNTs/CB particles during the friction process hinders the direct contact between the rubber and the metal, lowering the metal's wear. When the content of CF was 5 phr, the CB particles and MWCNTs inside the rubber were completely adsorbed by CF. The CFs within the rubber matrix are also synergistic with each other, and at this time, the filler form within the rubber presents a perfect state. A complete spatial network structure is formed within the rubber matrix. On the one hand, the spatial network structure formed by the synergistic CF/MWCNTs/CB promotes the dispersion of CB particles. It reduces the CB particle agglomerates within the rubber matrix. On the other hand, the spatial network structure hinders the direct contact between rubber and metal, resulting in less metal wear. When the content of CF within the rubber matrix exceeds 5phr, the strong interaction force between CF makes the CF appear in the rubber matrix with a serious agglomeration phenomenon. The agglomerates of CF also hinder the dispersion of CB particles, increasing the agglomerates of CB particles inside the rubber matrix, which substantially increases the unevenness of the rubber surface. At the same time, the agglomerates of CF severely hinder the spillage of aromatic oil from the CB particles. This all adds to the increasing wear of the metal.

As can be seen from Figure 11, the metal surface roughness all increased after friction. The change of metal surface roughness is mainly related to the number of particle agglomerates within the rubber matrix and the oil film, and the presence of the oil film can effectively reduce the amount of metal surface roughness change.

There is no obstruction of aromatic oil in CB particles by CF in CF-free rubber, and the fragrant oil spills out of the rubber surface to form a complete oil film. The presence of the oil film effectively reduces the amount of roughness variation on the metal surface. However, the lack of adsorption of CF on CB particles resulted in the presence of more CB particle agglomerates within the rubber matrix. The CB particle agglomerates make the rubber surface uneven, which causes severe fluctuations during the friction process. Therefore, despite the presence of oil film, the amount of change in metal surface roughness after rubbing the blended rubber without CF addition with metal is still significant.

With the increase of CF content within the rubber matrix, synergistic interaction with CF and CB particles and MWCNTs occurs, and more and more CB particles and MWCNTs are adsorbed on the CF surface. This promotes the dispersion of CB particles and MWCNTs and helps to reduce the number of CB particle agglomerates within the rubber matrix. When the CF content is 5phr, the CFs entanglement, the CB particles, and MWCNTs inside the rubber synergize perfectly with the CFs, and the vast majority of CB particles and MWCNTs are adsorbed on the CF surface. There are only very few CB particle agglomerates at this time inside the rubber, and the filler form inside the rubber is a perfect spatial network structure. The spatial network structure formed by the synergistic CF/MWCNTs/CB particles hinders the direct contact between rubber and metal, all of which flattens the rubber surface and helps reduce the variation in metal surface roughness. At the same time, a small amount of CF prevents some of the aromatic oils from spilling out, but there is still a more significant amount of aromatic oils spilling out to form an oil film. This also reduces the amount of change in metal surface roughness. 

When the content of CF exceeds 5 phr, severe CF agglomeration occurs within the rubber matrix. The agglomerates of CF also hinder the dispersion of CB particles and MWCNTs, increasing particle agglomerates within the rubber matrix and decreasing the flatness of the rubber surface. At the same time, the agglomerates of CF seriously hinder the spillage of aromatic oil from CB particles and prevent the formation of the oil film. This all makes the amount of variation of metal surface roughness increase.

Question (17):·Pag 16. Raw 461. The authors claim: "When the CF content is 5phr, the CFs lap each other".

- The authors are invited to specify what do they mean with "lap each other";

Response to comments:

Thank you for your comments. I'm confusing you due to my misrepresentation. I have corrected it to "entanglement".

When the CF content is 5phr, the CFs entanglement, the CB particles, and MWCNTs inside the rubber synergize perfectly with the CFs, and the vast majority of CB particles and MWCNTs are adsorbed on the CF surface.

Question (18):·Pag. 16. Raw 469. The authors claim: "...a more significant amount of scented oil spilling...".

- The authors are invited to specify what do they mean with "scented oil";

Response to comments:

Thank you for your comments. I will explain it to you in detail.

Due to the quality of carbon black raw material oil and the production process control, some aromatic oils are adsorbed on the surface of carbon black products. These aromatic oils will form a thin oil film on the rubber surface.

Due to my misrepresentation, I should replace "scented oil" with "aromatic oils" in the text.

At the same time, a small amount of CF prevents some of the aromatic oils from spilling out, but there is still a more significant amount of aromatic oils spilling out to form an oil film. This also reduces the amount of change in metal surface roughness.  

Question (19):·Pag. 17. Raws 482 to 488. The authors claim: "The agglomerates of CF cause the spatial network structure inside the rubber matrix to stack, the spatial structure inside the rubber matrix to be disturbed, and the dispersion of CB particles to gradually decrease. The amount of wear between rubber and metal and the change in metal roughness gradually decreases during the mixing process. ".

- The authors are invited to rewrite the sentence, improving the English language, despite the related message could be understood and agreed;

Response to comments:

The agglomerates of CF make the spatial meshwork of the rubber stack and disorder, resulting in a gradual decrease in the dispersion of CB particles. In addition, the spatial meshwork co-constructed by CF/MWCNTs/CB could restrict the overflow of aromatic oil from the rubber surface, so that the wear between the rubber and metal and the change of metal roughness gradually decrease during the whole mixing process.

Question (20):·Pag. 17. Raw 494 to 495. The authors claim: "When the content of CF exceeds 5 phr, the spatial network structure starts to stack and disorder and increases with the increase of CF content..."

- Also in this case, the authors are invited to rewrite the sentence, improving the English language.

Response to comments:

However, when the amount of CF is more than 5 phr, the spatial meshwork begins to be stacked and disordered. As the amount of CF increases, the spatial meshwork would be more disordered, which makes the wear of metal and the change of metal roughness increase accordingly.
